# Optimized Xception Learning Model and XgBoost Classifier for Detection of Multiclass Chest Disease from X-ray Images

**DOI:** 10.3390/diagnostics13152583

**Published:** 2023-08-03

**Authors:** Kashif Shaheed, Qaisar Abbas, Ayyaz Hussain, Imran Qureshi

**Affiliations:** 1Department of Multimedia Systems, Faculty of Electronics, Telecommunication and Informatics, Gdansk University of Technology, 80-233 Gdansk, Poland; kashifshaheed1@gmail.com; 2College of Computer and Information Sciences, Imam Mohammad Ibn Saud Islamic University (IMSIU), Riyadh 11432, Saudi Arabia; iqureshi@imamu.edu.sa; 3Department of Computer Science, Quaid-i-Azam University, Islamabad 44000, Pakistan; ayyaz.hussain@qau.edu.pk

**Keywords:** computer-aided detection system, artificial intelligence, deep learning, chest X-ray, COVID-19 detection, pneumonia, viral infection, feature extraction, Xception, inception-V3

## Abstract

Computed tomography (CT) scans, or radiographic images, were used to aid in the early diagnosis of patients and detect normal and abnormal lung function in the human chest. However, the diagnosis of lungs infected with coronavirus disease 2019 (COVID-19) was made more accurately from CT scan data than from a swab test. This study uses human chest radiography pictures to identify and categorize normal lungs, lung opacities, COVID-19-infected lungs, and viral pneumonia (often called pneumonia). In the past, several CAD systems using image processing, ML/DL, and other forms of machine learning have been developed. However, those CAD systems did not provide a general solution, required huge hyper-parameters, and were computationally inefficient to process huge datasets. Moreover, the DL models required high computational complexity, which requires a huge memory cost, and the complexity of the experimental materials’ backgrounds, which makes it difficult to train an efficient model. To address these issues, we developed the Inception module, which was improved to recognize and detect four classes of Chest X-ray in this research by substituting the original convolutions with an architecture based on modified-Xception (m-Xception). In addition, the model incorporates depth-separable convolution layers within the convolution layer, interlinked by linear residuals. The model’s training utilized a two-stage transfer learning process to produce an effective model. Finally, we used the XgBoost classifier to recognize multiple classes of chest X-rays. To evaluate the m-Xception model, the 1095 dataset was converted using a data augmentation technique into 48,000 X-ray images, including 12,000 normal, 12,000 pneumonia, 12,000 COVID-19 images, and 12,000 lung opacity images. To balance these classes, we used a data augmentation technique. Using public datasets with three distinct train-test divisions (80–20%, 70–30%, and 60–40%) to evaluate our work, we attained an average of 96.5% accuracy, 96% F1 score, 96% recall, and 96% precision. A comparative analysis demonstrates that the m-Xception method outperforms comparable existing methods. The results of the experiments indicate that the proposed approach is intended to assist radiologists in better diagnosing different lung diseases.

## 1. Introduction

COVID-19 initially emerged in Wuhan, China, in late 2019, prompting substantial efforts to detect this ailment. However, the detection of COVID-19 by manual testing requires a lot of time, effort, and complexity [1]. Additionally, test kits are hardly distributed globally. The percentage of false negatives, on the other hand, fluctuates with the duration of the illness. The false-negative rate in [2] was 20% five days after the onset of symptoms, but it was substantially greater (up to 100%) early in the infection. Visual indices linked to COVID-19 can be seen on chest X-ray imaging [3]. Additionally, a quick, efficient, and reasonably priced diagnostic to detect COVID-19 infection is a chest X-ray scan [4]. Chest X-ray scans are available; however, an experienced radiologist is required to detect the COVID-19 infection. The world’s healthcare systems are already overburdened due to the enormous number of illnesses. For the automated diagnosis of COVID-19 infections and the distinction between them and other illnesses, artificial intelligence (AI) systems can offer an alternate approach [5].

Numerous AI systems have demonstrated their effectiveness in the analysis of medical pictures, including the diagnosis of pneumonia [6] and semantic segmentation [7]. Similar to this, other AI systems have been developed, and their performance in diagnosing lung infections from chest X-ray pictures has shown encouraging results [4,5,8,9,10]. Deep convolutional neural networks are more effective than hand-crafted techniques because they can immediately extract important and high-level characteristics from data [11]. The foundation of hand-crafted approaches is extracting characteristics from planned models [12]. Since COVID-19 first emerged, significant attempts have been made to identify the infection using X-ray images. Due to two major issues, this discipline has not made much progress in recognizing lung infection as a practical use. The lung X-ray scans’ initial issue is their technical limitations. The second issue is the lack of standardized protocols, classes, and datasets for training the model. Because each study in the literature specifies its own protocol, classes, and data, it is challenging [13] to compare several solutions. Our goal in this study is to coordinate activities in this area. We started by producing a large quantity of lung X-ray-augmented images. Additionally, we created two scenarios for separating lung scans from lung scans of other lung disorders in four classes of lung diseases, as shown in Figure 1. In order to provide better comparisons across various methodologies, we also established train/val/test divides.

We gathered the largest lung X-ray scan database from public sources. To distinguish between COVID-19, lung opacity, normal, and viral pneumonia infection, we preprocessed and augmented those images. The training, validation, and test sets were separated into separate databases for each. Many testing data classes came from fresh sources that were not utilized in developing the training and validation sets. We employed deep learning (DL) architectures to differentiate among four classes of lung diseases. We also suggested an ensemble technique based on the advanced DL model and XgBoost classifiers.

Recent studies have shown that deep learning-based models can classify and find plant diseases, even though they have many problems. In this study, we add new convolutional layers to the Xception module to help it recognize and diagnose diseases in the lungs. So, the already-trained ResNet was combined with the modified Xception (m-Xception) module to extract high-quality visual features of lung illness. The classification results are then made more stable by adding a logarithm to softmax to optimize features. Figure 1 depicts a graphic representation of chest X-ray scans.

### 1.1. Limitation and Main Contributions

In past studies, authors have mainly utilized chest X-ray images and multiple DL and pre-trained DL models to detect lung-related diseases. However, these methods only emphasized performance enhancement. Since there is no restricted access to publicly available data at the beginning of the pandemic, many models are confined by the volume of lung-infected diseases [14]. In addition, the majority of established approaches also use several deep learning (DL) models, which increases their computational complexity [15]. Additionally, many of these techniques focus on enhancing CXR image quality before using them with a DL-based model. It is not yet possible to obtain accurate chest X-ray pictures for the purpose of disease identification in the lungs through the application of a machine learning algorithm. Our proposed research concentrates on using CXR images to diagnose four-class lung-related diseases, including COVID-19. The primary objective of this study is to develop a straightforward method for detecting three classes of lung diseases using ML algorithms. To detect lung-related diseases, our study includes preprocessing, feature extraction, and classification steps.

The following are the contributions of our proposed work.

Data augmentation and preprocessing steps are performed to make a dataset balanced and enhance the regions of the lungs for a better feature extraction step.Current solutions rely on the Inception model, which is incapable of creating space behavior and extracting feature maps from noisy areas. This is due to the significant number of false positives generated by the model, which, in the end, diminishes the model’s overall effectiveness. This is taken care of by the Xception architecture, which, in the model that has been proposed, essentially pulls features from all noisy-level portions while also enhancing the performance of the model in comparison to the one that came before it.Xception Net suffers from an overfitting problem due to a lack of regularization, which produces flash results on an unseen dataset and causes model performance scores to decline. To overcome this problem, regularization is used in the proposed model to improve scores and show that log functions work perfectly. The m-Xception model incorporates depth-separable convolution layers within the convolution layer, interlinked by linear residuals.The proposed model employs the Xception architecture as the backbone. Features are extracted from the images, and then Xception with 2D convolutional layers manages the robust features. The features pass them as input into the last layer, where the XgBoost classifier recognizes them.Accuracy, precision, recall, and the F1 score were used to evaluate the performance of the results. In addition, a comparison was made between the proposed work and existing methodologies to classify lung-related diseases.

### 1.2. Paper Organization

The remaining sections of this paper are structured as follows: the related studies are described in Section 2. The proposed methodology is described in detail in Section 3. The tests that were carried out on the findings in Section 4 The discussion of this study is presented in Section 5. In the final part of this study, the findings are discussed in Section 6.

## 2. Methodological Background and Related Studies

To predict and categorize COVID-19 and lung-related diseases from radiography images, many artificial intelligence (AI) frameworks were developed through advanced machine learning (ML) techniques such as deep learning (DL). Compared to simple DL methods, the pre-trained models [16,17] outperformed based on the architecture of convolutional neural networks.

**VGG-19:** The VGG Network is a sophisticated neural network architecture that makes use of three convolutional layers with a variety of layered filters to improve image extraction capabilities. The addition of an extra convolutional layer to each of the three convolutional blocks in VGGNet-19′s three convolutional blocks distinguishes it from other VGG variations and gives it more depth than VGGNet-16. With this specific architectural modification to VGGNet-19, capacity is increased, and visual aspects may be represented in a more complex manner.

**EfficientNet:** CNN introduced EfficientNet to address its scalability issues. The width and depth scales of CNNs must be increased to improve their accuracy. However, this escalation increases the cost of training and assessment time.

**GoogLeNet:** GoogLeNet is composed of nine inception modules and twenty-two hidden layers. The inception modules allow for the selection of available filter sizes within each block. By creating three groups of inception modules and adding three objective functions to each group, the problems of overfitting and gradient vanishing are resolved.

**Xception:** Xception is a sophisticated CNN that takes the concept of inception to extremes. It could be considered an extension of the Inception architecture. It introduces novel inception layers that are generated by depthwise convolution layers and a point-wise convolution layer, respectively.

**InceptionV3:** nceptionV3 is a widely used deep learning model for computer vision tasks, particularly image classification. It is part of the Inception family of models, also known as GoogLeNet. InceptionV3 represents an advancement over its predecessors and has been designed to improve both accuracy and efficiency. It is a powerful deep learning model designed for image classification tasks. Its architecture, featuring Inception modules and reduction techniques, enables it to strike a balance between accuracy and computational efficiency. The model’s pretrained weights are valuable for transfer learning, making it a popular choice in the field of computer vision.

**ResNet 50:** ResNet is a cutting-edge neural network architecture that has revolutionized CNN architecture. This was accomplished by adding a special block design and selecting long-range connectivity, both of which improved the system’s performance. The models with a moderate architecture that consists of a Fifth Convolutional layer with pooling applied between the layers, along with a Flattened, Dropout, and Single Fully Connected Layer, won the ImageNet competition. A computer-aided diagnosis system based on deep learning has been suggested by the authors in references [15,16,17].

We describe a system that utilizes chest X-ray radiology imaging to classify patients into several classes, such as normal (healthy), pneumonia subjects, COVID-19, and virally infected patients. We have also employed several pre-trained Convolutional Neural Network (CNN) models, namely ResNet50, ResNet101, ResNet152, InceptionV3, and Inception-ResNetV2. These models were trained on publicly available datasets. Among all the models used, the pre-trained Xception and ResNet50 models achieved the highest rate of classification accuracy.

### Related Studies

On the basis of Chest X-Ray images, multiple CADs have been developed to automatically classify lung-related diseases. In the following paragraphs, a concise description of these CADs is provided.

The authors in [18] have introduced COVID-NET as the initial open-source deep neural network for COVID-19 case investigation via CXR scans. In their study, the authors claimed that they were able to attain an accuracy rate of 92.4% and a sensitivity rate of 80%. The COVID-NET has been successful, but it is a sophisticated network that uses a lot of memory, which has three main drawbacks: high cost and extended execution time. These elements could create problems with accessibility and availability. The effectiveness of three cutting-edge, pre-trained deep learning models, such as VGG, ResNet50, and Inception-V3, for classifying lung scans into COVID-19-positive and non-COVID-19 categories was then evaluated and compared.

The authors in [19] used four distinct TL deep CNN models: AlexNet, ResNet18, DenseNet201, and SqueezeNet. They used 5247 chest X-ray images consisting of bacterial, viral, and normal chest X-ray images, which were preprocessed and trained for a classification task based on TL. The authors of [20] have suggested a technique for multi-classifying CT scans into three categories (pneumonia, uninfected, and infected with COVID-19). They conducted research on transfer learning to the convolutional blocks of four popular ResNet versions using publicly available image collections, despite a shortage of training data. The pre-trained ResNet50 architecture outperformed other groups in COVID-19 identification, outperforming them with accuracy, specificity, and sensitivity values of 91.21%, 88.46%, and 94.87%, respectively.

On the dataset, we trained the Faster R-CNN, the Mask R-CNN [21], and the ResNet-50 models, three different CNN-based models. Our results show that the ResNet-50 and Mask R-CNN models had an accuracy of 83% and 72%, respectively, while the Faster R-CNN had the best accuracy at 87%. One approach for measuring error is through the utilization of the Error framework. However, a significant disadvantage of this method is the extensive duration required for training, which may take several hours, as stated by the authors. In the study [22], the EDL-COVID model was developed. By creating a cascade of various deep learning architectures and using only one training run to make predictions, the authors were able to cut down on computation costs and training time. However, similar models produced by this method limit the advantages of combining them for comparable forecasts and prediction errors. Researchers introduced a new hybrid model in [23], which used a ResNet50 architecture to obtain deep feature vectors and then fed them into the linear kernel function of an SVM classifier. The approach was evaluated based on classification accuracy and achieved an impressive 94.7% accuracy rate.

The majority of the currently available studies for the detection of COVID-19 infection from X-ray scans have employed deep learning approaches [23,24,25,26,27,28,29,30,31,32,33,34,35,36,37,38,39,40], which have been motivated by the success of these techniques in numerous computer vision tasks. Seven various CNN designs, including VGG-19 [27], DenseNet-121 [28], Inception-V3 [29], ResNet-V2 [30], InceptionResNet-V2 [31], Xception [32], and Google MobileNet-V2 [33], were evaluated. They use a binary categorization of positive and negative COVID-19 infections, and their database only comprises COVID-19 cases [34]. According to their findings, the DenseNet-121 and VGG-19 models performed the best.

Mangal et al. [35] used CheXNet [36], which had its training data from the ChestX-ray database [37]. To identify the COVID-19 infection in three- and four-class settings, they applied transfer learning. With an identification rate for COVID-19 infection equal to 90.5% in the three-class scenario, they obtained an encouraging outcome. In [38], however, the authors used two new datasets, which include three-class and five-class COVID-19 datasets. They evaluated various deep learning architectures for both databases. In addition, they created an Ensemble-CNN method that excels beyond deep learning architectures. In other words, the proposed Ensemble-CNNs performed very well in recognizing a COVID-19 infection, with a 100% success rate in the three-class scenario and a 98.1% success rate in the five-class situation. Table 1 shows the relevant work performance as well as its constraints.

The input picture is first classified as normal or aberrant using a decision tree. This decision tree’s accuracy rate is 98%. A second decision tree determines whether aberrant photos are classified based on whether they show COVID-19 symptoms or not. This is a considerable improvement over the collection of feature selection techniques and CNN designs. The state-of-the-art studies that have been mentioned, the databases that were used, and the outcomes are summarized in Table 1. This chart reveals that, with a small number of X-ray scans, particularly for the COVID-19 class, the databases utilized vary from one job to the next. Additionally, each work establishes distinct classifications and grading procedures. As a result, we compiled publicly available X-ray scans for COVID-19, made our own COVID-19 X-ray scans available, and established an evaluation procedure and set of possible outcomes.

## 3. Materials and Methods

Figure 2 below represents the overall systematic flow diagram of our proposed work to detect various types of lung diseases when diagnosed through X-ray images. In our research, we developed an improved version of the Inception module for the recognition and detection of four classes of Chest X-rays. Instead of using the original convolutions, we adopted an architecture based on modified-Xception (m-Xception), which utilizes depth-separable convolution layers [39] interlinked by linear residuals within the convolution layer. This innovative approach allowed us to efficiently capture information at different spatial scales while significantly reducing the model’s complexity. To create an effective model, we employed a two-stage transfer learning process, pre-training the m-Xception on a large dataset and fine-tuning it on the Chest X-ray dataset. The resulting model demonstrated exceptional performance in recognizing and detecting the target classes. Finally, we utilized the XgBoost classifier [40] as an ensemble method to further enhance the model’s multi-class classification capabilities. Our research represents a significant step forward in the field of Chest X-ray analysis [41], offering a promising approach for accurate and efficient medical image interpretation.

In this section, we describe in detail a few different approaches that serve as the foundation for the algorithm we have developed for the detection of lung diseases. The proposed approach is described in the subsequent paragraph.

### 3.1. Data Acquisition

To evaluate the efficacy of our suggested approach, we used two public datasets made available to us by Tawsifur-Rehman [42]. Researchers from Qatar University in Doha, Qatar, the University of Dhaka in Bangladesh, and their collaborators from Pakistan and Malaysia, as well as medical professionals, have created a valuable database through their collaboration. This database contains chest X-ray images of COVID-19-positive cases, as well as normal cases and cases of viral pneumonitis. The objective of the initiative is to facilitate medical imaging research, particularly in light of the COVID-19 pandemic [42]. A visual example of the dataset is displayed in Figure 3.

In both datasets, there are a total of 21,165 samples, which are separated into four primary categories: COVID-19, lung opacity, normal, and Viral pneumonia. We can see that the Mean, Max, and Min values vary based on the image class by separating the class as shown in Figure 4. Viral pneumonia is the only category whose distribution is Normal-like across all three analyses. The maximum value that can be assigned to an image is 255. As anticipated, this is where the majority of classes reach their maximum size. Viral pneumonia is the class with the highest proportion of samples with lower Max values than the others. The majority of samples fall in the 200–225 range. Normal (Healthy) and Lung capacity samples have mean value distributions that are extremely similar. This may or may not be related to the fact that these classes are the most prevalent in the dataset. Different maxima on the distribution may also be attributable to the image source (e.g., two distinct institutions). Concerning the Max values, lung opacity and COVID-19 have comparable distributions (see “bumps”), whereas Normal patients have peaks at 150 and 250.

### 3.2. Proposed Methodology

Despite traditional DL models producing excellent results, there are critical issues regarding the validity of TL or DL models because the COVID-19 database, which was compiled from a smaller set of instances, has a serious bias issue. Thus, the prediction outcomes of the DL model can be vulnerable to generalization errors and high variance due to the limited database and noise. Additionally, stochastic training approaches can be sensitive and produce diverse weights during each training process. To overcome these issues, we propose in this study applying the transfer learning method to an existing Xception model. Xception-Net models are characterized by high accuracy. The design of the base network, Xception-Net, is built on a residual block and a depthwise separable convolutional neural network. Xception Net is based on three flows. Figure 2 represents the architecture of the Xception-Net [39] model. We have integrated the ResNet model to solve the issue of bias for high performance while maintaining computational efficiency. In fact, the model incorporates depth-separable convolution layers within the convolution layer, interlinked by linear residuals. To make our network simple, we have used the first entry flow for the feature extraction task. For classification purposes, we have applied the XgBoost [40] classifier. Overall, m-Xception with the XgBoost classifier categorizes lung images into four classes: COVID-19, pneumonia, viral opacity, and normal. The workflow phases of the proposed technique are thoroughly detailed in the following subsections.

Algorithm 1 also describes these procedures.
**Algorithm 1.** Classification of X-ray chest diseases by using m-Xception and XgBoost classifierStep 1:Pre-process image = X and Pre-processing step is applied by using:(a) Resize Chest X-Ray image (X) to (299, 299)[Enhanced Chest X-Ray image by preprocessing steps](a) Remove Noise using Gaussian smoothing operator, and (b) Enhance local contrast logarithmic operatorStep 2:Class Balance = Augmentation (preprocessed)Step 3:[Extract Deep Feature]:(a) Feature Extraction used: Optimize the m-Inception model by first entry flow for the feature extractionStep 4:Deep-features = Deep features were extracted by the m-Xception modelStep 5:Prediction = XgBoost classifier is used to classify the images into four classes: lung opacity, COVID-19, pneumonia, and normalStep 6:[End]

#### 3.2.1. Pre-Processing Steps

The objective of the preprocessing phase is to identify and reduce the number of image noises that are presented in chest X-ray images. This phase is necessary for X-ray images since many radiological images have noise and undesired artefacts that need to be removed in order to effectively identify lung-related disorders like COVID-19. These artefacts include patient apparel and wiring. In our proposed method, we employ straightforward preprocessing steps, including image resizing, noise removal, and local contrast enhancement. Using a Gaussian smoothing filter, we removed the disturbance from X-ray images. A logarithm operator is then utilized to enhance the local contrast of an image. This logarithm operator is also known as the pixel logarithm operator because it increases the value of pixels with low intensity. Figure 5 displays the unaltered and enhanced images.

#### 3.2.2. Data Augmentation to Control Class Imbalance

The data augmentation step is necessary to generate the class balance dataset. This step is also used to avoid model overfitting. Typically, new enhanced chest X-ray pictures are created by combining geometric augmentations. The 1095 chest X-rays used in this study for the evaluation of our proposed m-Xception–XgBoost system were broken down as follows: 375 normal, 345 viral pneumonia, and 375 COVID-19. First, 10 percent, 20 percent, and 80 percent, respectively, of the data from the training, testing, and validation sets were used. Then, we used a data augmentation strategy to prevent class imbalance. As shown in Table 2, the 1095 dataset is converted using a data augmentation technique into 48,000 X-ray images, including 12,000 normal, 12,000 pneumonia, 12,000 COVID-19 images, and 12,000 lung opacity images. Figure 6 depicts geometric enhancements individually applied to an X-ray image. This illustrates the effect of each geometric augmentation method and provides a sense of their relevance to the reader.

#### 3.2.3. Proposed Model for Features Extraction

The Xception network is achieving image classification based on the Inception_V3 network [41]. In the Xception network, the original convolution in inception V3 is replaced by deep separable convolution, resulting in an increase in network width and a reduction in parameters and model calculations. Additionally, the network features a residual connection mechanism similar to the ResNet network, which accelerates convergence speed, enhances classification accuracy, and improves the learning ability of the network for detailed features. This approach improves the model’s performance without increasing network complexity, making it applicable to COVID-19 and normal image classification. The model incorporates depth-separable convolution layers within the convolution layer, interlinked by linear residuals. The initial flow is based on the Xception network and serves as the feature extraction network. The second flow, which has separate convolution layers, is the intermediate flow. The middle layer is repeated eight times. Exit flow is the final layer. It is the final layer, and it produces the dense layer.

We have attempted to train the convolution layer filters in three dimensions, encompassing the physical dimensions as well as the channel width and height. By using a single convolution kernel, we can map spatial and cross-channel correlations simultaneously. To enhance the effectiveness of this process, the Inception module has broken it down into a set of procedures that independently analyze cross-channel and spatial connections, resulting in a simplified and improved approach. To improve the efficiency of the procedure, the Inception module was developed based on the idea of examining cross-channel correlations using multiple 1 × 1 convolutions. This enables the transformation of input data into three or four smaller spaces, which are then subjected to common 3 × 3 or 5 × 5 convolutions as shown in Figure 7 to map all correlations in the smaller 2D regions.

**Depthwise separable convolutional Layer:** Chollet (2017) [39] proposed the use of depth-separable convolution, a technique that separates spatial and channel data, to significantly reduce irrelevant feature parameters. The depthwise convolution aims to obtain feature maps by using a 3 × 3 convolution kernel, while the Pointwise convolution is employed to relate the output features across the feature channels through a standard 1 × 1 convolution kernel. The residual connection links six separable convolution modules. The model of depth-separable convolution incorporates two independent convolutions and a Max-pooling layer. To extract features and fit training data, the ReLU activation function is utilized. Additionally, a 1 × 1 convoluted operation is applied to ensure uniformity of input and output data. This architecture is visually represented in Figure 8.

The proposed model differs from Normal Inception and Xception by performing an 8 × 8 convolution before any N × N convolution and not using an immediate ReLU function for non-linearity. The experiment shows improved results without an immediate ReLU function, despite the presence of residual connections in the architecture, which maintain high performance even with a change in the order of operations. This reduces complexity while still achieving similar performance levels to existing convolution settings.

In the implementation of the proposed Xception Net, the arrangement of internal layers is demonstrated through Figure 7, which displays the core order of convolution layers used. The n-dimensional input tensor undergoes the log(Softmax(x)) operation, which can be expressed in a simplified manner as Log SoftMax:(1)$$LogS\;of\;tmax=\frac{\log(\exp\left(x\right))}{\sum \exp(x)}=x-\log((\exp\left(x\right)\log(\sum \exp(x)))$$

The loss function of the proposed model is optimized by utilizing the logarithmic Softmax of variable *x*. Moreover, to avoid over-fitting during training, a Regularize function is implemented to regulate the layers.
(2)$$Loss=\frac{1}{n}\sum_{1}^{n}Li+ℷRw_{i}^{2}$$
where L = L × log(avg(R)).

#### 3.2.4. Formulation of the Classification Model

As described in this section, the feature classification portion of our proposed system differentiates between coronavirus2 and non-coronavirus2 instances. Based on the texture information acquired, this phase is predominantly responsible for classifying each chest X-ray image into three diagnostic classes (COVID-19, lung opacity, pneumonia, and normal). The classification module significantly depends on clinical diagnostic case availability. This collection of instances with an early diagnosis is referred to as the “training set”. The technique used is referred to as “supervised learning”. Numerous classification techniques are available in the literature for the recent task of COVID-19 classification.

XgBoost, a robust technique for both regression and classification, was developed by Chen and Guestrin [40]. XgBoost is a set of victorious programs derived from Kaggle machine learning contests. The framework is based on gradient boosting and enhances efficiency and performance by fitting residual values with iterative decision trees. Unlike Friedman’s gradient boosting, XgBoost uses a Taylor expansion to estimate the loss function, which results in a superior tradeoff between bias and variance. This often leads to higher accuracy with fewer decision trees. The image below shows how XgBoost is represented.

Assuming a sample set consisting of n samples and m features, it can be represented as $D=f\left(x_{i},y_{i}\right)\times g(D_{j})$, where x represents the eigenvalue, and y represents the actual value. The final predicted value is obtained by summing the results of K trees and is expressed as follows.
(3)$${y_{i}}^{*}=\sum_{k=1}^{n}f\left(x_{i}\right),{\;f}_{k}\epsilon \;F$$

*F* represents a set of decision trees, each of which is specified as follows:(4)$$F=\left\{f\left(x\right)=w_{q}(x)\right\}\left\{q:R^{m}\to T,\;w\;\epsilon R^{T}\right\}$$

In XgBoost, the value of $f\left(x\right)$ stands in for one of the trees, and $w_{q}(x)$ stands for the weight of the leaf nodes. *T* is the total number of leaf nodes, and *q* represents the arrangement of each tree that links a sample to its matching leaf node. As a result, the anticipated value of XgBoost is equal to the total of each tree’s leaf node values. This model aims to minimize the goal function below while learning these *k* trees.
(5)$$L^{t}=\sum_{i=1}^{n}I(y_{i},{y_{i}}^{*})+\sum_{k=1}^{n}\partial (f_{k})$$

Loss is the difference between the true value and the estimated value in statistical analysis. Different loss function types, including exponential, square, and logarithmic losses, are frequently employed. The decision tree penalty is determined using regularization W to avoid overfitting. The formula for W is as follows:(6)$$\partial \left(f_{k}\right)=\omega T+1/2\alpha {\left\|w\right\|}^{2}$$

In the regular term, *T* is the number of leaf nodes, and is regarded as a hyper-parameter controlling the model’s complexity. The penalty coefficient, represented by the letter, for the leaf weight *w* is typically a fixed amount. The complexity of the model is often determined by empirical assignment, and in order to accommodate the residuals from the prior round, a new tree is inserted during the training phase. Consequently, the model’s expression when t trees are included is as follows:(7)$$y_{i}^{t}=y_{i}^{t-1}+f_{k}(x_{i})$$

Substituting (6) into the objective function (4) yields the function as
(8)$$L^{t}=\sum_{i=1}^{n}{I(y_{i},{y_{i}}^{*}{)}^{t-1}\times f}_{t}\left(x_{i}\right)+\partial \left(f_{k}\right)$$

The objective function is then expanded using the Taylor method by XgBoost, which also removes the high-order small infinitesimal terms and takes the first three terms.
(9)$$L^{t}\approx \sum_{i=1}^{n}{I(y_{i},{y_{i}}^{*}{)}^{t-1}+g_{i}f_{i}\left(x_{i}\right)+1/2h_{i}f_{t}^{2}(x_{i})\times f}_{t}\left(x_{i}\right)+\partial \left(f_{k}\right)$$

The objective function must be optimized by taking into account the first and second derivatives of the loss function, denoted as *g_i_* and *h_i_*, respectively. Notably, the residual between the prediction score is deleted because it has no effect on the optimization procedure.
(10)$$L^{t}\approx \sum_{i=1}^{n}{g_{i}f_{i}\left(x_{i}\right)+1/2h_{i}f_{t}^{2}(x_{i})\times f}_{t}\left(x_{i}\right)+\partial \left(f_{k}\right)$$

The final objective function is obtained:(11)$$Objective-function=-1/2\sum_{j=1}^{T}\frac{G_{j}}{H_{j}+\mu }+\rho T$$

Regularization is generally added to the standard function via XgBoost, which reduces model complexity. The first and second derivatives are used to fit the residual error. This technique also permits column sampling to lessen computation and overfitting. Compared to gradient-boosting decision trees (GBDT), this generates more hyper-parameters, although it can be difficult to effectively tweak them. Hyper-parameter optimization is a useful approach to solving this problem, but it takes a lot of work, experience, and prior information on the part of the researchers.

## 4. Experimental Results

All of the experiments, analyses, and evaluations were conducted on a Dell laptop outfitted with an Intel(R) Core(TM) i7 processor, 16 GB of RAM, a 1.50 GHz processor, and Microsoft Windows 10 x64. Using the Adam optimizer, the focal loss function = 2, and a batch size of 64, all networks were trained for 10 to 30 iterations using the Adam optimizer. The learning rate was 1 × 10^−6^ for the first 10 epochs and 1 × 10^−7^ for the next 30 epochs. Active data augmentation was used to normalize, resize, and crop the input images in order to achieve the correct input dimensions for each network. The input image dimensions for the network were 299 × 299 pixels.

### 4.1. Performance Evaluation Metrics

The m-Xception model’s efficacy is measured using a variety of quantitative evaluation criteria such as sensitivity, specificity, F1-score, accuracy, recall, and precision. False metrics are derived from four primary indicators: the proportion of healthy cases correctly classified (true positives, TP), the proportion of healthy cases wrongly classified (false negatives, FN), the proportion of unhealthy instances correctly classified (true negatives, TN), and the proportion of unhealthy cases mistakenly categorized (false positives, FP).

Accuracy, Recall, Specificity, and F-score are some metrics used as evaluation indicators to determine how well various models perform. These metrics were derived from the statistics of correct and incorrect detections. To put this into a context that is easier to understand, these measurements and their enlarged computations for multi-class activities that use macro averages are written out as Equations (12)–(15).
(12)$$\mathrm{Accuracy}=\frac{TP+TN}{TP+TN+FP+FN}$$
(13)$$\mathrm{Precision}=\frac{TP}{TP+FP}$$
(14)$$\mathrm{Recall}=\frac{TP}{TP+FN}$$
(15)$$F-\mathrm{score}=\frac{2\times \mathrm{Prcision}\times \mathrm{Recall}}{\mathrm{Precision}+\mathrm{Recall}}=\frac{2\times TP}{2\times TP+FP+FN}$$

AUC can also be referred to as “Area Under the Curve”. It provides efficacy measurements across all thresholds. A model with 100% inaccurate predictions has an AUC of 0.0, whereas a model with 100% accurate predictions has an AUC of 1.0.

### 4.2. Results Analysis

Several experiments were performed to evaluate the proposed preprocessing steps. Those experiments were more about the performance analysis of the proposed m-Xception and XgBoost models compared to other state-of-the-art studies found in the literature. We conducted several experiments by utilizing three transfer-learning-based CNN architectures, such as ResNeXt-50, Inception-v3, VGG19, and DenseNet-161, and our proposed m-Inception technique for the recognition of chest X-ray diseases. In general, the comparisons were performed on general TL architectures along with state-of-the-art approaches based on implementation models. Those TL models were used in the corresponding papers, and we used the same hyper-parameters as used in our m-Xception model.

We utilized the validation and testing splits to assess the effectiveness of these procedures.

Due to the high-level nature of the original Xception‘s architecture, such time disparities were to be anticipated. Inception-v3 is a 22-layer convolutional network, whereas our m-Xception model has only three convolutional layers along with dense layers. However, the most important conclusion from this comparison is not about accuracy, but about training. Training time is the quantity of time required to train over 12,000 images. The comparison clearly demonstrates how quick Inception is despite its structural complexity. On a CPU, it performs better than the simplest feasible architecture. Inception has effectively utilized the sparse connectivity attribute of convolutional networks and the combination of numerous sparse connections to represent a dense structure, which has ultimately led to speedier learning. Even though there is no mention of training time in the base paper, we can infer that their CNN model would take longer to train than Inception-v3 because their model is structurally more complex. Therefore, if we were to execute the m-Xception model, it would take 10 S for prediction on our CPU, and training would take longer.

Figure 9 depicts the training and testing accuracy with loss for the proposed transfer learning (TL) model. The investigation was conducted using the 10-fold cross-validation method. For each fold, the training accuracy (Acc), testing accuracy (Val Acc), training loss (Loss), and testing loss (Val Loss) were measured for the training and testing samples.

The confusion matrix of each class of chest X-ray is represented in Figure 10. The confusion metrics, where it marginally outperformed our suggested technique, are shown in this figure, which shows the findings of the testing data. The testing data’s confusion matrices are shown in Figure 10. The key finding is that the proposed system correctly classified COVID-19 samples with a 100% success rate. The distinction between the normal and pneumonia classes caused the most confusion for all models. Since every model correctly identified COVID-19 sample 100% of the time, we counted the samples that were incorrectly categorized as COVID-19 for the testing split, as shown in Table 3 and Table 4, with respect to different settings of hyper-parameters. According to this table, the proposed system, which had the fewest false positives, was the best model. Based on the findings, we deduce that the categorization of the four chest X-ray classes and the identification of COVID-19 are more stable using our suggested technique. In addition, we have also drawn the AUC curve for the proposed system in terms of data augmentation techniques with respect to four different classes without preprocessing steps. This AUC curve shows that it is necessary to perform data augmentation to accurately determine the four classes of chest X-ray, as shown in Figure 11.

Table 3, Table 4 and Table 5 compare the four tested models for class recognition for each of the different classes. These tables show that our suggested method is the most effective in identifying two of the four classifications. These tables provided a clearer explanation for the identification of the various classes. We can see from these confusion matrices that the proposed strategy performs best (98.1%) for identifying COVID-19 samples. In addition, all models (the best of which is proposed at 98.1%) accurately identify the lung opacity, and there are no pneumonia samples. This occurred because there was only one source for the lung opacity and no pneumonia class samples (we only discovered one source for this class).

As previously mentioned, it is essential to know how the network extracts features from preprocessed chest X-ray images. Using Score-CAM-based heat maps for the proposed m-Xception system, which were created from preprocessed images and disease-detected regions from images. In fact, it is possible to ensure that the classification algorithm can learn which affected regions of the images are significant, and which are not. The heat map of the detected lung regions and the original lungs demonstrates (Figure 12) that they are related to the four classes of lung diseases. When the m-Xception model with the proposed system employs normal chest X-ray images to categorize lung diseases, regions are not always the most significant portions of the images. Figure 12 shows, however, that the m-Xception system is compatible with X-ray images, which is advantageous for a biomedical application of this significance. Alternatively, the disease-detected regions of images are superior for the CAD system because they make it simpler to classify diseases using chest X-ray images.

As can be seen in Table 3, Table 4, Table 5 and Table 6, our proposed model achieves virtually perfect predictive accuracy across three distinct training-test splits. When recognizing COVID-19 occurrences from test pictures (a chest X-ray), the AUC compares the true positive rate against the false positive rate to determine the suggested classifier’s superior classification performance. As a result, the proposed effort is justified in being suggested as a viable option for identifying COVID-19 situations.

Table 6 displays the overall comparisons with state-of-the-art approaches such as Hemdan [28], Apostolopoulos [27], Edoardo [38], Mangal [35], Yoo [15], and Turkoglu [3]. These state-of-the-art systems are compared with our proposed m-Xception architecture with the same hyper-parameter settings and preprocessing steps. Compared to other systems, the proposed m-Xception model outperforms them. As shown in Figure 13, we have also compared the loss versus accuracy of relevant studies when measured over 25 epochs. Comparison graphs for validation and loss for training and testing state-of-the-art systems such as Hemdan [28], Apostolopoulos [27], and Edoardo [38] are shown in Figure 13.

The suggested diagnostic approach was compared to related investigations conducted in the past [25,34,35,36,37,38]. The suggested diagnostic system’s superiority to the existing detection methods was shown using a number of estimation measures. Table 5 summarizes the quantitative comparisons of state-of-the-art systems. The F1-score, precision, recall, and average accuracy of our suggested CAD model, which is based on three separate train-test divisions, are all above 96%. Table 5 shows that our proposed method has better recall, precision, F1-score, and average precision than prior studies with similar designs. Therefore, our proposed CAD system may prove to be an invaluable asset to hospitals in their efforts to diagnose COVID-19 cases. Figure 13 depicts the training and testing accuracy with loss for the proposed transfer learning (TL) model. The investigation was conducted utilizing the 10-fold cross-validation method. For each fold, the training accuracy (Acc), testing accuracy (Val Acc), training loss (Loss), and testing loss (Val Loss) were measured for the training and testing samples.

The proposed m-Xception model with preprocessing and data augmentation steps was compared to comparable other studies [25,34,35,36,37,38] conducted previously. Several estimation metrics were used to demonstrate that the proposed diagnostic system is superior to the current detection schemes. Table 6 and Table 7 summarize the quantitative comparisons in terms of different statistical metrics. We have observed that our m-Xception model, based on three distinct train-test divisions, achieves a high F1-score, precision, recall, and average accuracy of 97%, 96%, 96%, and 97%, respectively. Consequently, our proposed CAD system could be a significant addition to healthcare facilities for COVID-19 case diagnosis.

The total time required to calculate the presented CAD solution for COVID-19, including picture preparation and the proposed technique, is around 2 s. This means that it can be used for instantaneous processing. The suggested approach also requires less processing power for the preliminary steps, the extraction of real-time texture features, and the subsequent categorization.

## 5. Discussion

Due to the lack of a cure, the COVID-19 virus has produced a massive outbreak. It is tough to treat since the disease mutates over time and has a single strand of RNA. COVID-19 has killed thousands of individuals, primarily in the United States, Spain, India, Italy, China, the United Kingdom, Iran, and other countries. “COVID-19 strains are found in humans, cats, dogs, hogs, chickens, rats, and other animals. Compared to COVID-19 lung disease, viral pneumonia and lung opacity symptoms include hoarseness, fever, headache, runny nose, and cough. Those with weakened immune systems are especially vulnerable to this potentially lethal infection. Globally, this communicable COVID-19 illness spreads quickly from person to person. Physical contact, respiratory contact, hand contact, and mucous contact are the most common ways that this virus can be transmitted from one individual to another. This member of the virus family is known to cause severe respiratory problems. Severe lung damage and acute respiratory pain are also possible outcomes in such cases [10]. As of 15 July 2020, there were 12,964,899 affected individuals worldwide, with 570,279 fatalities. Current evidence suggests that individuals with chronic health conditions and the elderly are more susceptible to the COVID-19 mortality rate. The virus is transmitted between individuals through coughing, sneezing, and respiratory secretions [34]. This virus frequently causes fever, inflammation, respiratory abnormalities, and illnesses such as pneumonia, multiple organ failure, and death [36,41]. Expensive and time-consuming laboratory investigations necessitate the use of a properly equipped research facility”.

A deep learning-based system is utilized to look for COVID-19 in CT scans. Multiple researchers [15,16] have created and made available to the public chest X-ray images of patients with COVID-19. COVID-19 is diagnosed using the COVID-Net technique [21] with these publicly available datasets. The results of using deep learning to diagnose chest X-ray images were promising. Deep learning models are frequently utilized to process medical imaging data. In Ref. [16], pneumonia detection is accomplished with the aid of convolutional neural networks. In this research, we present a deep-network-based automated technique for COVID-19 diagnosis. The multiresolution analytic capability is used in the proposed network.

By model, the photographs are separated into three categories: normal, viral pneumonia, and COVID-19. In addition, a comparative analysis is conducted to assess the effectiveness of the proposed method. The proposed method can be used to diagnose COVID-19 based on chest X-ray images. Utilizing X-ray images will assist in disease management. Figure 6 shows the performance of the proposed model per epoch; the left graph shows loss reduction as the number of epochs increases, and higher values indicate that the losses are decreasing. The right-side curve shows accuracy for training and validation; based on the above curves, it is evident that the model can overcome overfitting as the curves for accuracy for both training and validation move in the same rhythm.

Table 3, Table 4 and Table 5 compare the proposed model’s performance on selected data with the performance of known solutions, depending on the number of epochs. Epochs refer to the number of iterations followed by training performed across the whole network. On both 10 and 50 epochs, the models included in this table were analyzed and compared to the proposed model. Accuracy in training and validation and losses incurred during training and validation were utilized as metrics to evaluate the proposed model. It was noted from the table that the suggested model outperforms existing solutions in terms of accuracy and losses. This was found because the proposed model improved its accuracy while simultaneously reducing its losses when 10 epochs were considered.

Table 6 displays the overall comparisons with state-of-the-art approaches such as Hemdan [28], Apostolopoulos [27], Edoardo [38], Mangal [35], Yoo [15], and Turkoglu [3]. These state-of-the-art systems are compared with our proposed m-Xception architecture with the same hyper-parameter sittings and preprocessing steps. Compared to other systems, the proposed m-Xception model outperforms them.

Table 6 compares the performance of the proposed models with that of existing models on a local dataset, which allows for robust testing of the proposed models with existing models on another dataset where existing ones may not be tested, on a different number of epochs, as results were compared when the number of epochs was 10 and when it was 30, as well as training, validation accuracy, and losses during the proposed model’s training. It was observed from the table that the proposed models achieve an average of 7% improvement over the existing solutions at epoch 10 on the local dataset for both accuracy and loss metrics. For 30 epochs of experiments, the proposed model performed around 6% better in terms of accuracy than the existing solutions. Table 6 explains the results for various classes and displays predicted and actual numbers and performance metrics such as accuracy and loss to evaluate model performance across classes. It was observed from the table that the ratio of correct to incorrect averaging had a stable gap, which means the model was not biased towards any particular class, which suggests that the model was stable towards multiple variations of datasets even when some subsets of classes were experimenting with the proposed model. In addition, the confusion matrix score for detecting pneumonia and normal lung disease is visually represented in Figure 14, and a visual diagram of the predicted system is shown in Figure 15.

### 5.1. Advantages of Current Study

Our extensive research endeavor focuses on employing chest X-ray images for the diagnosis of COVID-19, normal, viral pneumonia, and lung opacity. In this paper, we have identified four classes of lung diseases. To detect lung diseases, we have utilized preprocessing, feature extraction through the m-Xception model, and classification through the XgBoost classifier. The contributions made by our proposed task are detailed below.
(1)To extract desirable features from enhanced chest X-ray images, we have devised a novel framework based on a convolution vision transformer and a linear residual model.(2)The XgBoost classifier was used to predict four-class lung diseases such as COVID-19, viral pneumonia, lung opacity, and normal cases using X-Ray images and a variety of train-test split techniques.(3)Performance metrics such as accuracy, precision, recall, and the F1 score were used to analyze the results. Furthermore, the planned study was compared to similar earlier work for the diagnosis of different lung disorders.

### 5.2. Limitations and Challenges of Current Study

Despite the fact that the performance of high-performing networks cannot be generalized to real-world applications, recent research suggests that deep learning (DL) models can make judgements based on irrelevant input [24,34,35,36]. Unlike traditional X-rays, the divided lungs made it easier for CNN to pinpoint the main Region of Interest (ROI). In other words, the end-user’s trust in the effectiveness of artificial intelligence (AI) must grow as a result of the network’s consistent classification decisions. Instead of using the entire chest X-Ray image to draw conclusions about lung diseases, the lung area should be used. Due to the scarcity of ground-truth mask measurements, the results presented using standard X-rays may fail wholly or partially in practical applications. Several authors developed a standard for lung hoods with the assistance of a team of radiologists. However, they are inadequate for training the model.

We were unable to perform the accuracy comparison on faster processing units because we lacked the processing capacity to do so. That would have allowed us to utilize hyperparameters, which would have allowed us to adjust the learning rates, processing volumes, etc. We believe that additional experiments with hyperparameters would have led to greater precision.However, if these parameters are computed using the CPU as opposed to the GPU, the process can take days.Nearly all image enhancement methods incorrectly categorized chest X-rays in a sample case as normal, viral pneumonia, lung opacity, or normal. Gamma enhancement surpasses other enhancement methods, which is an interesting observation. As depicted in Figure 13, the Grade-Cam score indicates that the proposed TL model outperformed and clearly made a difference among lung diseases. In conclusion, this study’s detection performance for COVID-19 and other lung infections (Table 3, Table 4, Table 5 and Table 6) is consistent with that reported in recent literature. However, this research adds context that has been lacking in other recent studies. Furthermore, no article has ever reported results utilizing such massive chest X-Ray images before. Since the models in this work were trained and validated using a sizable dataset, the obtained findings are competitive with state-of-the-art methods, trustworthy, and applicable beyond the scope of the current study.An FPGA-based implementation [43] of the described model can provide performance boosts in terms of faster inference times, hardware acceleration, reduced power consumption, and optimized resource usage. However, it requires expertise in FPGA programming and careful consideration of cost and resource constraints. The potential advantages of FPGA-based implementations are particularly attractive for applications with real-time processing needs or resource-constrained environments. However, this is not primarily concerned with this research. This point of view will be addressed in future applications of this proposed model.This paper used the Xception TL model as the backbone of the architecture. However, the Xception model is an improvement over InceptionV3 in several aspects. Both models are based on the concept of “Inception” modules, which use multiple filters of different sizes to capture features at various scales. However, Xception improves upon InceptionV3 by introducing depthwise separable convolutions, resulting in better efficiency, improved representation learning, and a smaller model size, while maintaining or even surpassing the performance of InceptionV3. Therefore, the implementation of the m-Xception model should be tested on an application in a resource-constrained environment.

## 6. Conclusions

This work proposes a cost-effective diagnostic scheme to identify COVID-19, viral pneumonia, lung opacity, and normal cases using chest X-ray images. The proposed method comprises three steps: data augmentation and preprocessing, feature extraction using the m-Xception-residual model, and classification through the XgBoost classifier. Existing solutions use Inception as a classification module with mobile net, and to extract features in the proposed model, they use a logarithmic-based softmax layer. These changes significantly reduce false-positive responses, which ultimately increase true positives, which are reflected in the accuracy figures of the proposed model. In addition, the local experimental materials consist of photographs of rice plants taken in realistic settings of wild fields, complete with various backgrounds and varying degrees of illumination intensity. Despite this, the experiment produced an acceptable conclusion, confirming the usefulness and practicality of the suggested technique for diagnosing crop diseases. Even though the circumstances of the setting were difficult, the experiment nevertheless had the potential to result in a valuable finding. Even if the model has very high accuracy and memory efficiency, its performance can still be further improved by performing additional fine-tuning operations on the backbone network. This is because the model’s performance is directly related to the amount of memory it uses. One way to accomplish this goal is to perform more activities on the backbone network. We have great expectations that, in the near future, we will be able to considerably increase the performance of the model while at the same time lowering its size. This goal will be attained by unfreezing more layers of bottom convolutional layers and fine-tuning the network architecture.

## Figures and Tables

**Figure 1 diagnostics-13-02583-f001:**
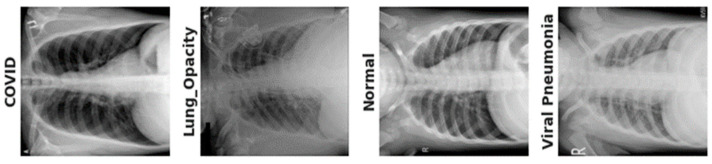
A visual diagram of lung-related diseases such as COVID-19, lung opacity, normal, and viral pneumonia infections.

**Figure 2 diagnostics-13-02583-f002:**
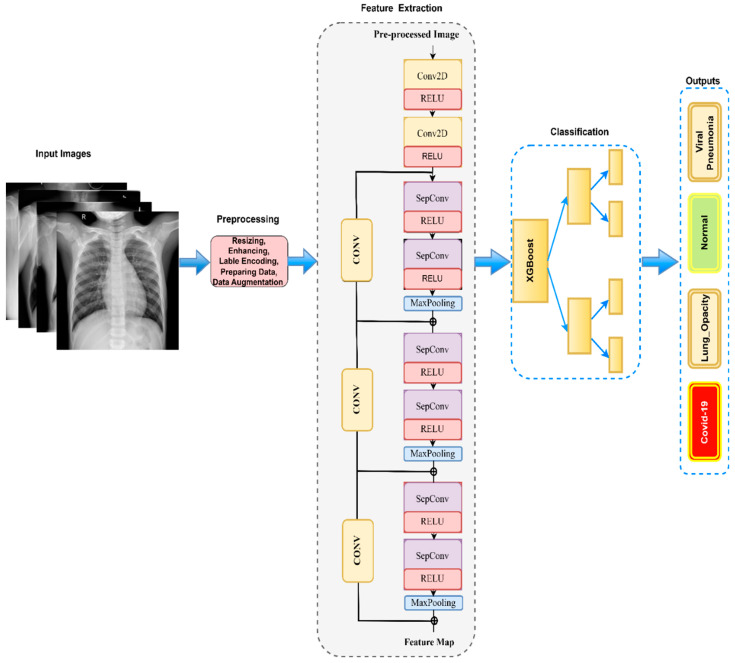
The proposed m-Xception transfer learning model’s systematic flow diagram, where Conv2d or Conv: Convolutional, RELU: Rectified linear unit, SepConv: Separable convolutional, and XGBoost: Extreme gradient boosting.

**Figure 3 diagnostics-13-02583-f003:**
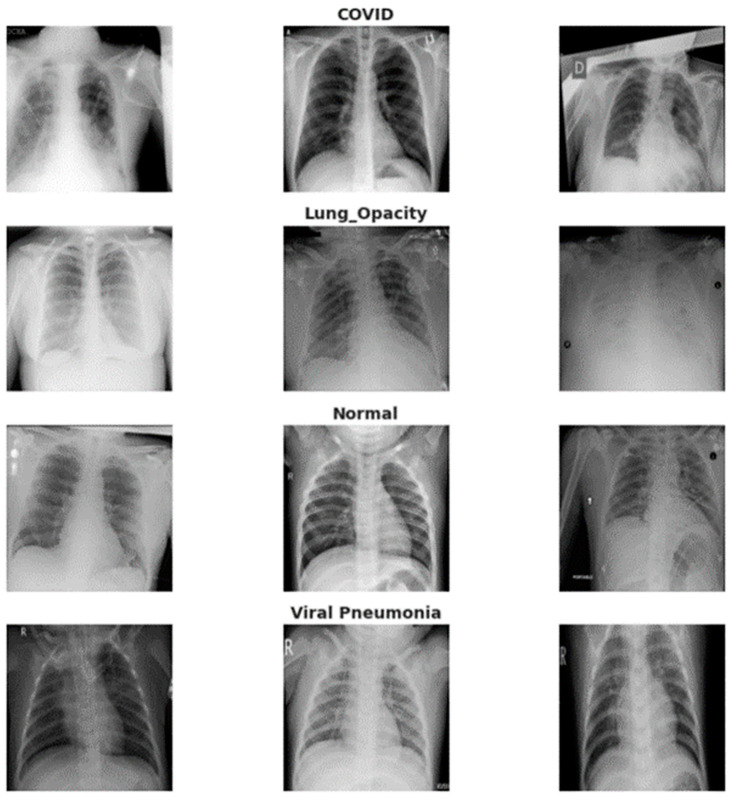
Normal and COVID-19 radiograph image samples from the Kaggle dataset.

**Figure 4 diagnostics-13-02583-f004:**
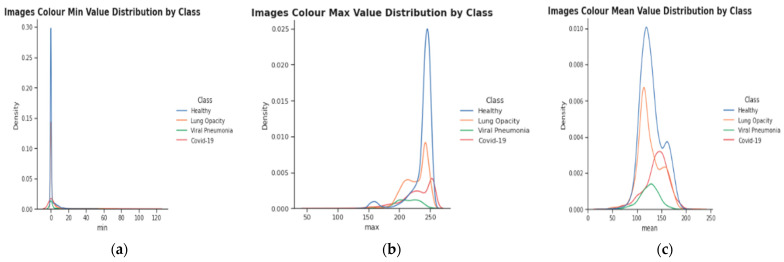
Representation of the separation by class; (**a**–**c**) Min, Max, and Mean values varying according to the image class, respectively.

**Figure 5 diagnostics-13-02583-f005:**
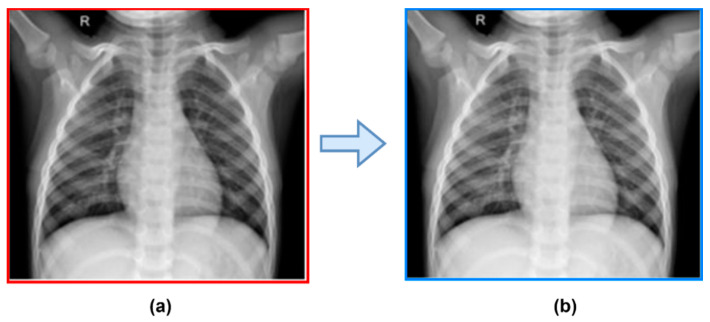
Image with contrast augmentation and smoothing: (**a**) raw image and (**b**) enhanced image.

**Figure 6 diagnostics-13-02583-f006:**
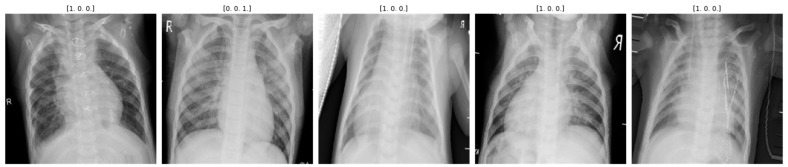
Some instances of data augmentation approaches for class balance.

**Figure 7 diagnostics-13-02583-f007:**
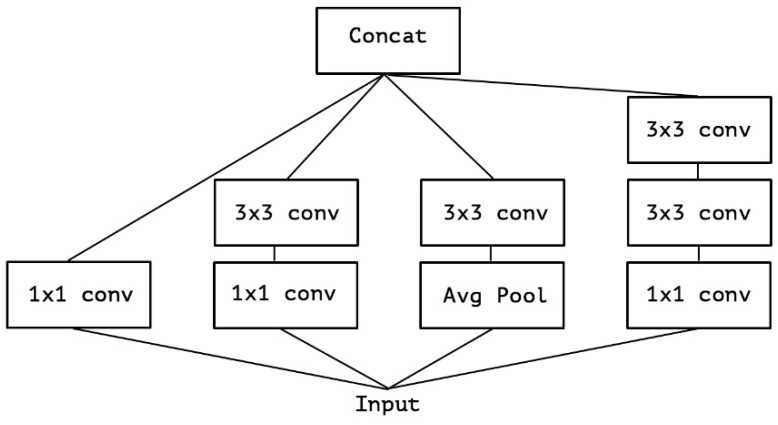
The canonical Inception architecture.

**Figure 8 diagnostics-13-02583-f008:**
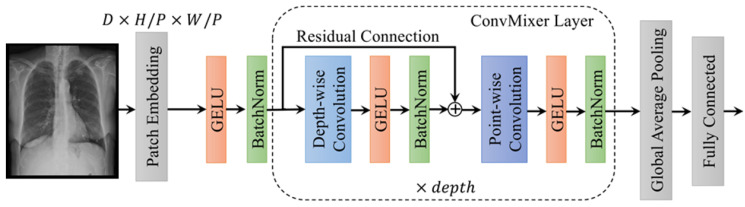
The convolution architecture utilized in this paper to the modified Xception (m-Xception) DL model, where GELU: Gaussian error linear unit, BatchNorm: Batch normalization, H, W: Kernel size (width and height), P: kernels and D:size of input tensor.

**Figure 9 diagnostics-13-02583-f009:**
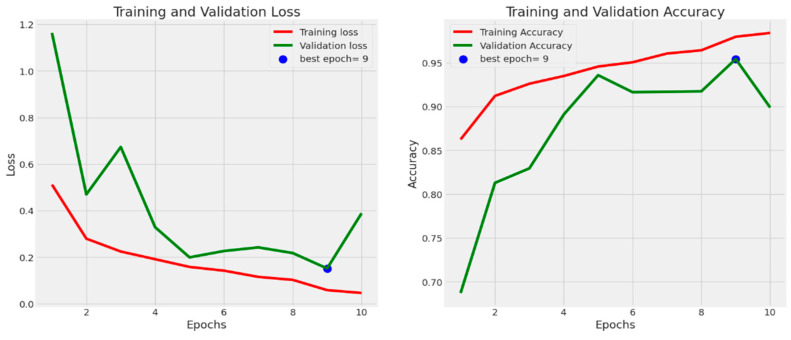
Model performance over accuracy/loss of the proposed architecture.

**Figure 10 diagnostics-13-02583-f010:**
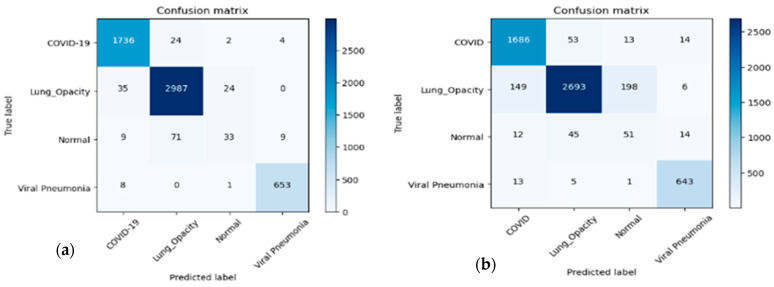
The confusion matrix of the proposed m-Xception system in terms of (**a**) data augmentation and (**b**) without data augmentation techniques on the dataset.

**Figure 11 diagnostics-13-02583-f011:**
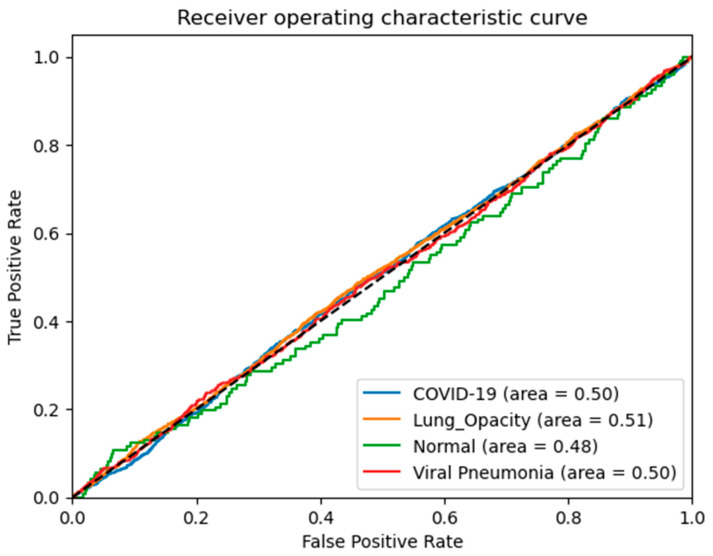
AUC curve on the original dataset without data augmentation techniques with respect to four different classes without preprocessing steps by the proposed m-Xception model.

**Figure 12 diagnostics-13-02583-f012:**
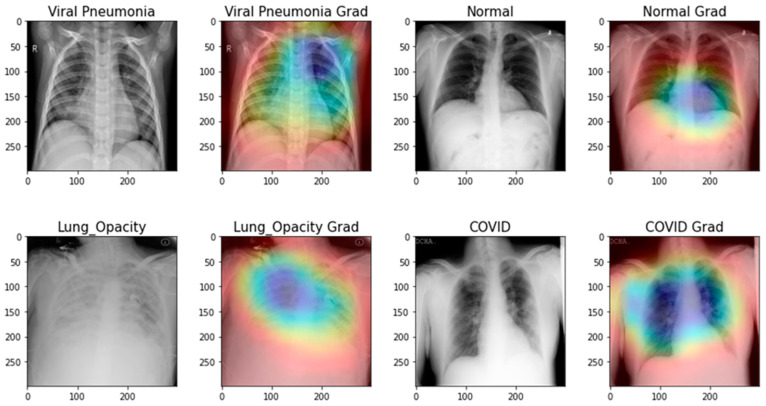
A Grad-cam-based analysis of the proposed m-Xception system to perform decision capabilities based on chest X-ray diseases (lung opacity, COVID, viral pneumonia, and normal).

**Figure 13 diagnostics-13-02583-f013:**
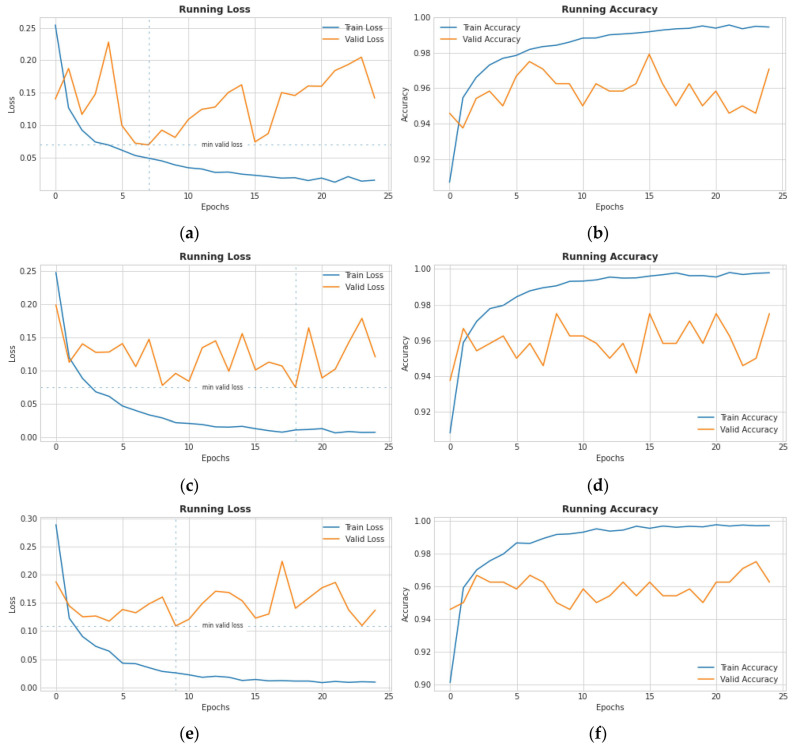
Comparison graphs for validation and loss for training and testing state-of-the-art systems such as (**a**,**b**) Hemdan [28], (**c**,**d**) Apostolopoulos [27], and (**e**,**f**) Edoardo [38].

**Figure 14 diagnostics-13-02583-f014:**
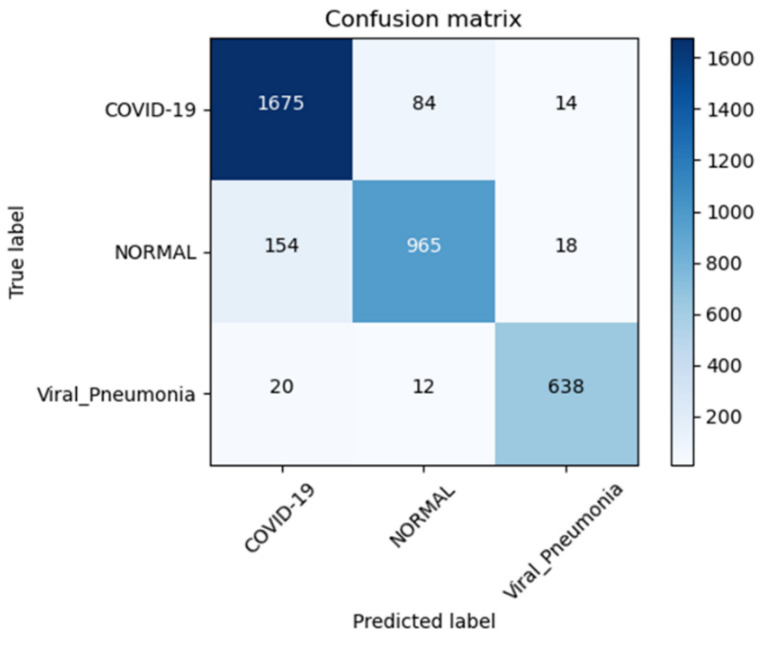
Confusion matrix for three-class (viral pneumonia, normal, and COVID-19) by the proposed m-Xception model with data augmentation and preprocessing steps.

**Figure 15 diagnostics-13-02583-f015:**
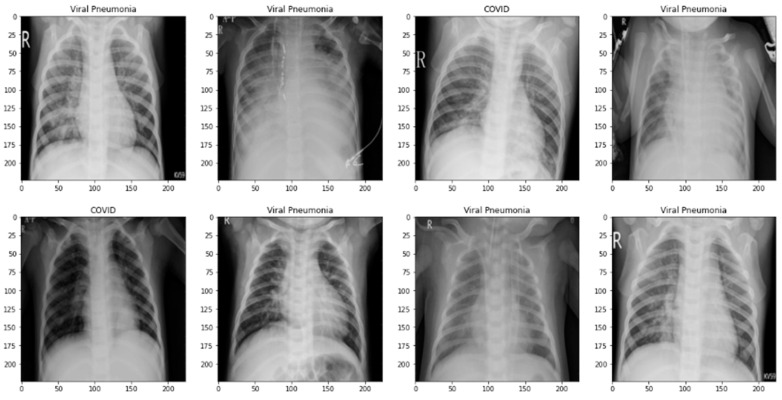
A visual diagram of the predicted results by proposed the m-Xception model.

**Table 1 diagnostics-13-02583-t001:** An overview of related research towards the detection of COVID-19.

Approach	Models	Class	* Output	Criterion	Constraints
Hemdan [28], 2020	VGG-19 ResNet-v2 DenseNet-201	COVID-19 and Normal	ACC = 88%, F1-score = 84%, SEN = 96%, SP = 81%	4.1 million trainable	Complex image preprocessing and classification stages render the method computationally challenging. Two divisions are recognized.
Apostolopoulos [27], 2020	VGG-19 MobileNet-v2 Inception Xception Inception-ResNet v2	COVID-19, pneumonia (bacterial, viral, and normal)	ACC = 98%, F1-score = 96%, SEN = 95%, SP = 95%	5.8 million trainable	A small number of COVID-19 samples used but five-classes of lung diseases were identified.
Edoardo [38], 2021	ResNetXt-50, Inception-v3, DenseNet-161	COVID-19, viral pneumonia, bacterial pneumonia, lung opacity, normal	ACC = 81%	12.00 million trainable	Only focused on lung infections without preprocessing and classify COVID-19 patients. Ensemble approach to obtaining multi-class label infection labels.
Mangal [35], 2020	COVIDAID	COVID-19, pneumonia and normal	ACC = 93%	11.78 million	The method has a degraded performance and is based on three classes only and no generalized solution.
Yoo et al. [5], 2020	AXIR 1 to 4	Normal, abnormal, TB, non-TB and COVID-19	ACC = 90%	NS	Training complexity several deep learning models was used for training.
Turkoglu [3], 2020	COVIDetectioNet	COVID-19, pneumonia and Normal	ACC = 99%. SEN = 100%, SP = 98%	NS	Non-effective evaluation of deep learning model because only one train-test split strategy is used.
Rehman et.al [19]	AlexNet, ResNet18, DenseNet201, and SqueezeNet	CT Images	ACC = 93%,	23 million	Limited in terms of data used in the work. No class imbalance and no preprocessing used.
Nasiri et al. [20]	ReseNet-50	Pneumonia, uninfected, and infected with COVID-19	ACC = 98%.	NS	Limited to three classes only, no preprocessing and data augmentation.

* ACC: Accuracy, SEN: Sensitivity, SP: Specificity and NIL: Not specified.

**Table 2 diagnostics-13-02583-t002:** Images included in each class of the COVID-19 database with and without data augmentation.

Category	Images	Data Augmentation
Normal	375	12,000
Pneumonia	345	12,000
COVID-19	375	12,000
Lung Opacity	400	12,000
Total	1495	48,000

**Table 3 diagnostics-13-02583-t003:** Proposed chest X-ray results with respect to four classes based on 10 epochs and a 20–80% test–train ratio.

Pathology	Accuracy	Precision	Recall	F1-Score
COVID-19	97.56%	95.33%	95.34%	97.16%
Lung Opacity	94.82%	89.33%	94.64%	94.38%
Normal	96.22%	96.67%	90.23%	95.80%
Viral Pneumonia	94.13%	95.33%	96.33%	97.33%

**Table 4 diagnostics-13-02583-t004:** Proposed chest X-ray results with respect to four classes based on 10 epochs and a 30–70% test–train ratio.

Pathology	Accuracy	Precision	Recall	F1-Score
COVID-19	97.56%	95.33%	95.34%	97.16%
Lung Opacity	94.82%	89.33%	94.64%	94.38%
Normal	96.22%	96.67%	90.23%	95.80%
Viral Pneumonia	94.13%	95.33%	96.33%	97.33%

**Table 5 diagnostics-13-02583-t005:** Proposed results chest X-ray with respect to four classes based on 10 epochs and a 40–60% test–train ratio.

Pathology	Accuracy	Precision	Recall	F1-Score
COVID-19	97.23%	95.00%	95.00%	97.26%
Lung Opacity	94.20%	89.11%	94.20%	94.08%
Normal	96.20%	96.20%	90.10%	95.10%
Viral Pneumonia	94.10%	95.13%	96.00%	97.11%

**Table 6 diagnostics-13-02583-t006:** Training, validation, and loss performance of state-of-the-art approaches.

Models/Methods	10 Epochs				30 Epochs			
	Training Acc. %	Validation Acc. %	Training Losses	Validation Losses	Training Acc. %	Validation Acc. %	Training Losses	Validation Losses
Hemdan [28]	75.6	75.4	88.14	88.41	73	71	83	82
Apostolopoulos [27]	82	82	76.44	73	76	75	85	81
Edoardo [38]	83.24	84	73.1	71	81	79	81	75
Mangal [35]	87.1	87	74	77	80	72	73	80
Yoo [15]	88.54	88	89	82	81	78	77	75
Turkoglu [3]	94.11	95.32	71	73	82	81	79	72
m-Xception	96.78	95	62.1	55	92	91	61	56

ACC: Accuracy.

**Table 7 diagnostics-13-02583-t007:** Training performance by different deep learning models.

Pre-Trained Models	10 Epochs				30 Epochs			
	Training Acc. %	Validation Acc. %	Training Losses	Validation Losses	Training Acc. %	Validation Acc. %	Training Losses	Validation Losses
VGG-19 [15]	75	74	55	52	67	70.5	70.3	72.5
DenseNet-121 [16]	77	75	52	53	74	78.1	70	70.4
Inception-V3 [17]	78	82	50	52	81	83.9	65.2	62.5
ResNet-V2 [18]	80	81	51	55	84	85.2	61.9	60.3
InceptionResNet-V2 [19]	83	84	50	50	89	88.9	59.4	59
Xception [20]	84	87	45	50	90.6	90	58	53.1
m-Xception	96	95	32	45	96.82	97.4	53.1	52

## Data Availability

We have used a publicly available dataset shared by Tawsifur Rehman [42]. This dataset can be downloaded at https://www.kaggle.com/tawsifurrahman/COVID19-radiography-database (accessed on 20 July 2023), https://github.com/Edo2610/COVID-19_X-ray_Two-proposed-Databases (accessed on 20 July 2023).

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
