# Peer review of "Optimized Xception Learning Model and XgBoost Classifier for Detection of Multiclass Chest Disease from X-ray Images"

_diagnostics, 2023, doi:10.3390/diagnostics13152583_

Round 1
Reviewer 1 Report
I want to clarify few doubts and would like to suggest some comments. These are,
1. Is this research improves the Optimize Inception Learning Model and XgBoost Classifier or predicting more accuracy of Chest Disease?
2. What is the benefit of taking Multiclass X-ray images? How do you experimented the different classes?
3. In the abstract, you have mentioned 12000 Chest X-rays were used. But how many number of COVID images, Lung_Opacity, Normal and Pneumonia ?
4. Page no 266-270, the provided data sets are not matching with 12000 images. Could you check this?
5. Improve the redability of Figure 4. Representation of the Separating by class, figure (a), (b) and (c) can visualize that the Mean, 286 Max and Min values vary according to the image class.
6.Equations are not cited in appropriate places.
7. Figure 10. Confusion matrix of proposed system- Where is the input data used?
8. Figure 11. AUC curve without data augmentation techniques-What is the unit value of X and Y axis?
Author Response
Original Manuscript ID: ID: diagnostics-2519823
Original Article Title: Optimize Xception Learning Model and XgBoost Classifier for Detection of Multiclass Chest Disease from X-Ray Images
To: Editor in Chief,
MDPI, Diagnostics
Re: Response to reviewers
Dear Editor,
Many thanks for insightful comments and suggestions of the referees. Thank you for allowing a resubmission of our manuscript, with an opportunity to address the reviewers’ comments.
We are uploading (a) our point-by-point response to the comments (below) (response to reviewers), (b) an updated manuscript with yellow highlighting indicating changes, and (c) a clean updated manuscript without highlights (PDF main document).
By following reviewers’ comments, we made substantial modifications in our paper to improve its clarity, English and readability. In our revised paper, we represent the improved manuscript such as:
(1) Revised Abstract, (2) Revised Introduction, (3) Results section, (4) Discussions and Conclusion sections.
We have made the following modifications as desired by the reviewers:
Best regards,
Corresponding Author,
Dr. Qaisar Abbas (On behalf of authors),
Professor.

Reviewer 2 Report
work is good for Society.
FPGA implementations of the proposed algorithms boost performance.
Author Response

(The authors gave the same response as above.)

Reviewer 3 Report
This study proposes an Inception module called m-Xception for identifying and categorizing normal lungs, lung opacities, COVID-19-infected lungs, and viral pneumonia using chest X-ray images. The model utilizes depth-separable convolution layers and linear residuals.
Some important points have to be clarified or justified and few concerns need to be addressed by the authors for the betterment of the manuscript
1- What is the main objective of this research paper?
2- It is suggested to include the following article in the related work:-
- Sergio Saponara, Abdussalam Elhanashi, Alessio Gagliardi, "Reconstruct fingerprint images using deep learning and sparse autoencoder algorithms," Proc. SPIE 11736, Real-Time Image Processing and Deep Learning 2021, 1173603 (12 April 2021); https://doi.org/10.1117/12.2585707
-
3- Can you elaborate on the data acquisition process, including the sources of the datasets used and their characteristics?
4- What were the steps taken to address class imbalance in the dataset, and how effective was the data augmentation technique in achieving this?
5- Explain the architecture of the proposed model and how it incorporates depth-separable convolution layers and linear residuals.
6- How was the performance of the proposed model evaluated, and what were the metrics used for the evaluation?
7- What were the results of the experiments conducted with the proposed models, and how do they compare to other state-of-the-art studies in the literature?
8- Were there any limitations or challenges encountered during the research, and how were they addressed?
9- Can you provide a detailed explanation of the process used for the visualization of the model's predicted output with Gradient-weighted Class Activation Mapping (Grad-CAM)?
Further proofreading is required
Author Response

(The authors gave the same response as above.)

Round 2
Reviewer 3 Report
Thanks to authors for improving the paper